# Research

evolution, palaeontology, taxonomy and systematics

invertebrate palaeontology, Eutardigrada, Miocene, *Paradoryphoribius*

**Authors for correspondence:**
Marc A. Mapalo
e-mail: marcmapalo@g.harvard.edu
Javier Ortega-Hernández
e-mail: jortegahernandez@fas.harvard.edu
Phillip Barden
e-mail: barden@njit.edu

Electronic material is available online at https://doi.org/10.6084/m9.figshare.c.5638782.

# A tardigrade in Dominican amber

Marc A. Mapalo[1], Ninon Robin[2], Brendon E. Boudinot[3,4], Javier Ortega-Hernández[1] and Phillip Barden[5,6]

[1]Museum of Comparative Zoology, Department of Organismic and Evolutionary Biology, Harvard University, 26 Oxford Street, Cambridge, MA 02138, USA
[2]Directorate Earth and History of Life, Royal Belgian Institute of Natural Sciences, Brussels, Belgium
[3]Institut für Spezielle Zoologie und Evolutionsforschung, Friedrich-Schiller-Universität Jena, Jena, Germany
[4]University of California, Davis, Department of Entomology, One Shields Avenue, Davis 94596, CA, USA
[5]Department of Biological Sciences, New Jersey Institute of Technology, Newark, USA
[6]Division of Invertebrate Zoology, American Museum of Natural History, New York City, USA

 MAM, 0000-0001-7653-0508; NR, 0000-0002-1756-3171; BEB, 0000-0002-4588-0430; JO-H, 0000-0002-6801-7373; PB, 0000-0001-6277-320X

Tardigrades are a diverse group of charismatic microscopic invertebrates that are best known for their ability to survive extreme conditions. Despite their long evolutionary history and global distribution in both aquatic and terrestrial environments, the tardigrade fossil record is exceedingly sparse. Molecular clocks estimate that tardigrades diverged from other panarthropod lineages before the Cambrian, but only two definitive crown-group representatives have been described to date, both from Cretaceous fossil deposits in North America. Here, we report a third fossil tardigrade from Miocene age Dominican amber. *Paradoryphoribius chronocaribbeus* gen. et sp. nov. is the first unambiguous fossil representative of the diverse superfamily Isohypsibioidea, as well as the first tardigrade fossil described from the Cenozoic. We propose that the patchy tardigrade fossil record can be explained by the preferential preservation of these microinvertebrates as amber inclusions, coupled with the scarcity of fossiliferous amber deposits before the Cretaceous.

## 1. Introduction

Fossils have an important role in reconstructing the history of complex life through deep time. However, some organisms are sparsely represented in the rock record, which severely hinders our understanding of their evolution. A notorious example are the tardigrades—also known as water bears or moss piglets—a charismatic group of microscopic invertebrates that are famous for their survival after exposure to extreme conditions, such as the vacuum of space and ionizing radiation [1]. Owing to their microscopic size and lack of heavily biomineralized body parts, the tardigrade fossil record currently consists of two described species regarded as members of the crown-group [2,3] and one Cambrian representative that may belong to the stem lineage [4,5].

Chronologically, *Beorn leggi* represents the first fossil tardigrade to be described, back in 1964 [2]. It is embedded in Canadian amber (chemawanite) from secondary deposits along Cedar Lake, Manitoba and dates to the Upper Cretaceous (*ca* 78 Ma). However, the lack of high-resolution images of taxonomically important characters, such as claw morphology, did not allow the placement of this taxon within any extant tardigrade families, and thus the new family Beornidae was erected to accommodate it [2]. Its precise affinities remain contentious, but some authors suggest it could belong within superfamilies Isohypsibioidea or Macrobiotoidea [3]. The amber piece containing *Beo. leggi* hosts a second, smaller individual that was originally regarded as a putative heterotardigrade. This smaller specimen is curled and shrivelled, which complicates a complete morphological description, so it remains formally unnamed and its affinities unresolved [2]. Almost four decades later, the fossil tardigrade *Milnesium swolenskyi* was described from New Jersey amber that is

stratigraphically attributed to the Turonian (Upper Cretaceous), making it approximately 14 Myr older than *Beo. leggi*. The preserved morphology of *Mil. swolenskyi* allowed for an unequivocal assignment to the Family Milnesiidae [6]. In fact, *Mil. swolenskyi* closely resembles extant *Milnesium* species in terms of overall body shape, presence of six oral papillae and claw morphology (*Milnesium*-type: with primary and secondary claw branches completely separated), indicating that the external cuticular morphology of this tardigrade group has remained largely unchanged for at least 92 Myr [3].

The stratigraphically oldest putative fossil tardigrade consists of four phosphatized specimens with Orsten-type preservation that were recovered from the middle Cambrian Kuonamka Formation in Siberia. All known specimens are attributed to a single taxon interpreted as a possible stem-group tardigrade [4]. The unnamed taxon has a tardigrade-like body outline and size, but greatly differs from extant representatives in the lack of the fully developed fourth pair of legs, different orientation of the legs, and claw morphology. The presence of papillae-like structures around the mouth and pillar-like cuticular structures on the ventral side of the body, in addition to the overall appearance and size, have been used to support the affinity of this enigmatic fossil as an ancestral tardigrade, and possibly a juvenile stage rather than an adult. Although some authors have proposed formalizing this taxon within Tardigrada [5], others emphasize the usage of stem-group description to account for the differences between these fossils and extant tardigrades [7].

In this study, we describe a crown-group tardigrade embedded in amber from the Dominican Republic and dated to the Miocene (approx. 16 Ma). The preserved morphology allows us to erect the new genus and species, *Paradoryphoribius chronocaribbeus* gen. et sp. nov., and assign it to the extant superfamily Isohypsibioidea. *Paradoryphoribius chronocaribbeus* gen. et sp. nov. represents the third unequivocal crown-group tardigrade in the fossil record described to date, and the first definitive fossil member of the diverse superfamily Isohypsibioidea. We discuss the importance of this discovery for understanding the preservation patterns of tardigrades in the fossil record and its contribution in providing temporal information of major evolutionary events of this enigmatic metazoan phylum.

## 2. Material and methods

### (a) Microscopy and imaging

We imaged the specimen through transmitted light and confocal fluorescence microscopy. For transmitted light microscopy, the specimen was mounted to a slide with dental wax and imaged under stereo microscopy with a Nikon SMZ25 auto montage system and Nikon ECLIPSE Ts2 inverted microscope. For the fluorescence microscopy, the specimen was prepared by putting glycerine (Immersol G, Zeiss) at both sides of the field of view. Autofluorescence of the cuticular structures was detected at an excitation wavelength of 488 nm using the LSM 700 Confocal Microscope (Zeiss) for the entire fossil and using the LSM 980 Confocal Microscope with Airyscan 2 detector (Zeiss) to obtain higher resolution images of taxonomically important characters, such as the claws and bucco-pharyngeal apparatus. Different optical sections were obtained to create the final image. Colour-coded projections of the optical sections were generated using Fiji 2.0 with the 'physics' LUT colour scheme. Inverted greyscale projections were also generated to highlight autofluorescence signals. The 'maximum intensity' projection type was used for both image reconstructions. The lighting properties of the images were adjusted using Adobe LIGHTROOM 9.3. Figures were assembled using Adobe ILLUSTRATOR 24.2.1. Schematic drawings were made using Adobe ILLUSTRATOR 24.2.1.

### (b) Morphometric measurement

All measurements are given in micrometres (µm). Body length was measured from the most anterior tip of the body to the most caudal part (except for the hind legs). Claws were measured according to Beasley *et al.* [8] to obtain the lengths of the primary claw branch, secondary claw branch and basal section. The *br* ratio or the ratio of the secondary claw branch length to the primary claw branch length was also measured [9]. Structures were measured using FIJI 2.0.

### (c) Phylogenetic analyses

To test the placement of the fossil relative to extant tardigrade superfamilies, we performed phylogenetic analyses using 28 morphological characters that can be grouped into four sets: body surface, claws, bucco-pharyngeal apparatus and egg morphology (electronic supplementary material, file S1). Because the goal of the analyses was to assess the superfamily affinity of the fossil, only type genera of all the 13 extant families within the four eutardigrade superfamilies (Eohypsibioidea, Hypsibioidea, Isohypsibioidea and Macrobiotoidea) were used, together with the fossil and *Milnesium tardigradum* as an outgroup to Parachela following the molecular-based tree topology in [10,11]. Character coding was based on the type species of the genera used. When possible, studies that provide the most recent redescriptions of the different species and high-quality microscope images were used as references for determining the character states. If not available, references of the original species descriptions were used, together with references containing high-quality microscope images of the type species, and monographs of descriptions of eutardigrade genera (for the full list of references used, see the electronic supplementary material, file S1).

The character matrix (electronic supplementary material, file S2) was then optimized under both maximum parsimony (MP) and Bayesian inference (BI). Parsimony searches were run in TNT 1.5 [12] under New Technology Search, using Driven Search with Sectorial Search, Ratchet, Drift and Tree fusing options activated with standard settings [13,14]. The analysis was set to find the minimum tree length 100 times and to collapse trees after each search. All characters were treated as unordered. For comparative purposes, analyses were performed under equal and implied weights ($k = 3$). Clade supports were calculated using jackknife ($p = 25$) and symmetric resampling ($p = 15$) [15] for the equal and implied weights analyses, respectively, with 1000 replicates each.

The Bayesian analysis was done in MRBAYES 3.2 [16] using the Mk model [17] + Gamma with the coding set to 'variable', which excluded two invariant characters. The analysis was run for 2 000 000 generations sampling every 500 generations and with an initial burn-in of 1000 samples (25% relative burn-in). Two runs were simultaneously done with each having one cold and three heated chains. Convergence was assessed by checking that the average deviation of split frequencies of the two runs were less than 0.01, effective sample size values were greater than 200 and the potential scale reduction factor was approximately = 1.

## 3. Results

### (a) Systematic palaeontology

Phylum Tardigrada Doyère, 1840 [18].
Class Eutardigrada Richters, 1926 [19].
Order Parachela Schuster, Nelson, Grigarick and Christenberry, 1980 [20].

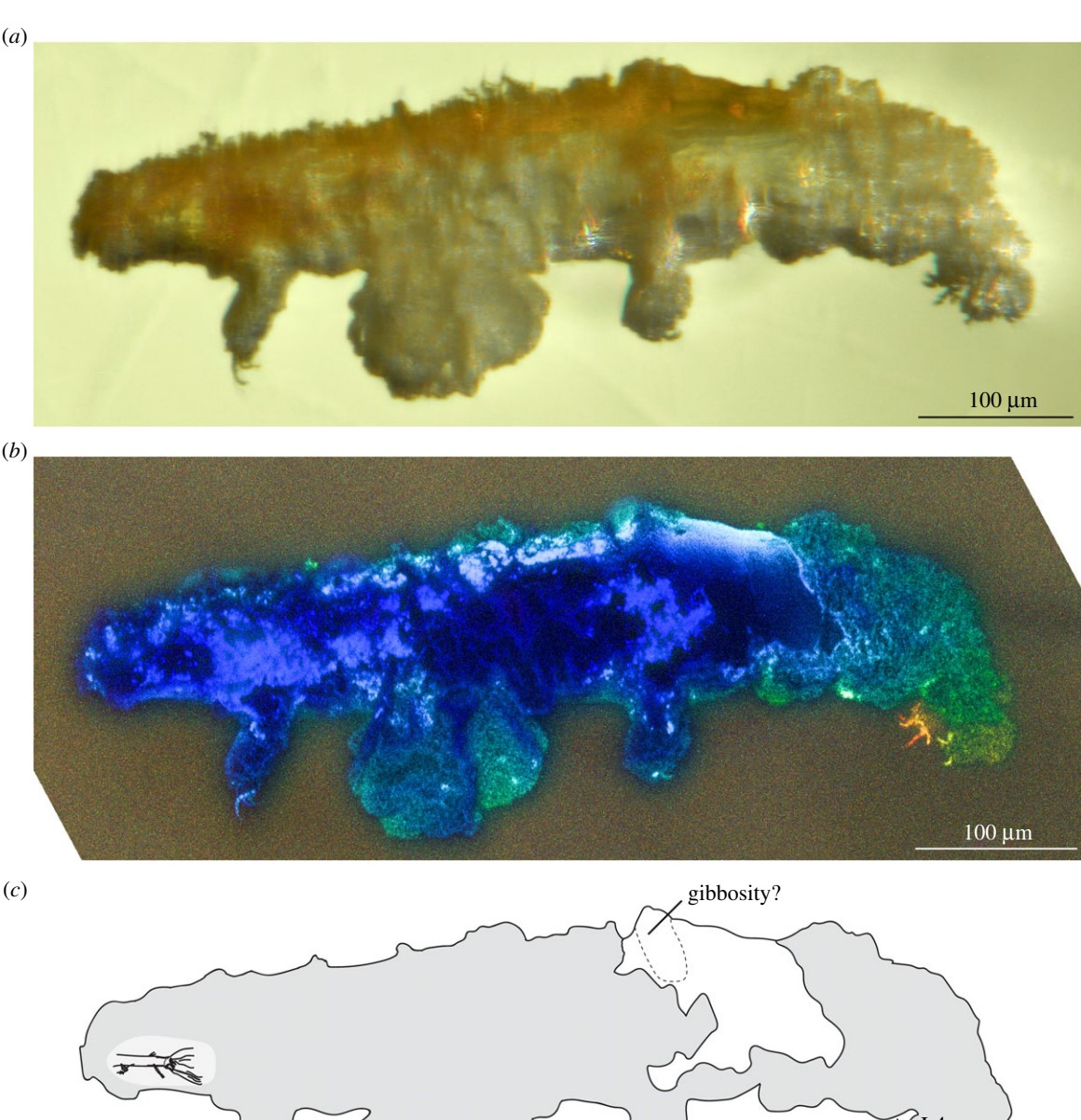

**Figure 1.** Left lateral view of *Paradoryphoribius chronocaribbeus* gen. et sp. nov. from Miocene Dominican amber. (*a*) Specimen photographed with transmitted light under stereomicroscope. (*b*) Specimen photographed with autofluorescence under confocal microscope at 488 nm; different colours indicate *z*-depth, with violet to red gradient representing the shallowest to deepest planes, respectively. (*c*) Schematic drawing; grey shading indicates rough-textured layer. Ln, leg number. (Online version in colour.)

Superfamily Isohypsibioidea Sands, McInnes, Marley, Goodal-Copestake, Convey and Linse, 2008 [21].
Genus *Paradoryphoribius* gen. nov. (Three letter acronym: *Pdo.*).

*Etymology*: owing to the close resemblance (para-) to the extant genus *Doryphoribius* [9].

*Diagnosis*: tardigrade with *Isohypsibius*-type claws (i.e. the basal section and secondary branch form a right angle) with the claw pairs slightly different in shape and size. Accessory points present but not clearly visible. Cuticular bar present between claws of the fourth pair of legs. Pseudolunules absent. Bucco-pharyngeal apparatus consists of a rigid buccal tube with a ventral lamina (ventral apophysis) for the apophyses of the stylet muscle insertion (AISM). No dorsal AISM observed. Pharyngeal apophyses and one thin macroplacoid present, but microplacoids appear absent. Cuticle smooth. Cuticular gibbosities (i.e. cuticular protuberances) may be present.

*Differential diagnosis:* by the presence of *Isohypsibius*-type claws and ventral lamina, the new genus is morphologically similar to *Doryphoribius* but differs in the presence of a single thin macroplacoid instead of separated granular-shaped macroplacoids present in *Doryphoribius*.

*Remarks*: the new tardigrade from Dominican amber (figures 1 and 2) can be placed within the superfamily Isohypsibioidea based on claw morphology, specifically the presence of *Isohypsibius*-type claws (figure 3). Even though the presence of ventral lamina and gibbosities indicate that the fossil is comparable to the genus *Doryphoribius*, we did not place the new genus within Doryphoribiidae owing to the current polyphyletic status of the different members of the family [9], as well as its different macroplacoid morphology (figure 4). The absence of the granular-shaped macroplacoids in *Paradoryphoribius* can either be a real biological signal or a taphonomic

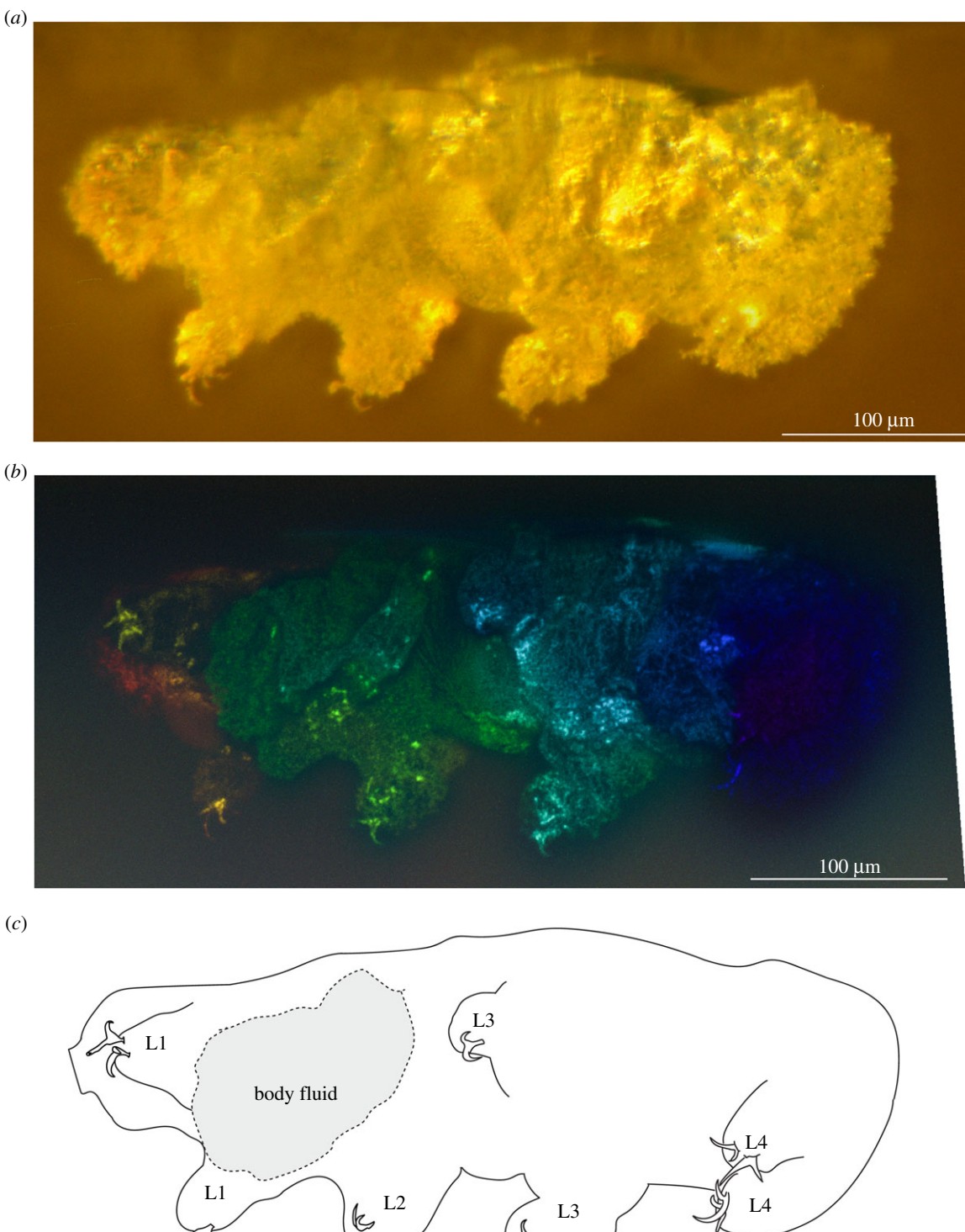

**Figure 2.** Ventral view of *Paradoryphoribius chronocaribbeus* gen. et sp. nov. from Miocene Dominican amber. (*a*) Specimen photographed with transmitted light under stereomicroscope. (*b*) Specimen photographed with autofluorescence under confocal microscope at 488 nm; different colours indicate *z*-depth, with violet to red gradient representing the shallowest to deepest planes, respectively. (*c*) Schematic drawing. Ln, leg number. (Online version in colour.)

artefact. Macroplacoids, when present, are visible under auto-fluorescence whether the bucco-pharyngeal apparatus is inside the organism or removed from the body [22–24]. In fact, macroplacoids can provide a stronger autofluorescent signal compared to other parts of the bucco-pharyngeal apparatus [23]. Because several regions of the bucco-pharyngeal apparatus were imaged in high-resolution (e.g. buccal tube, pharyngeal apophyses and stylet support; figure 4c–e), we consider this as support for the observed morphology of the macroplacoid of *Paradoryphoribius*. Single undivided macroplacoids are not observed in any genera of Isohypsi-bioidea but are found in some genera of the subfamily Itaquasconinae (Hypsibioidea: Hypsibiidae) [6,25]. This simi-larity of the macroplacoid shape can be explained by high frequency of homoplasy of the bucco-pharyngeal apparatus observed in extant eutardigrades [9,26]. Lastly, the clustering of *Paradoryphoribius* together with other genera of the superfam-ily Isohypsibioidea in the phylogenetic analyses supports its taxonomic placement within this extant group (figure 5).

*Type species*: *Paradoryphoribius chronocaribbeus* sp. nov.

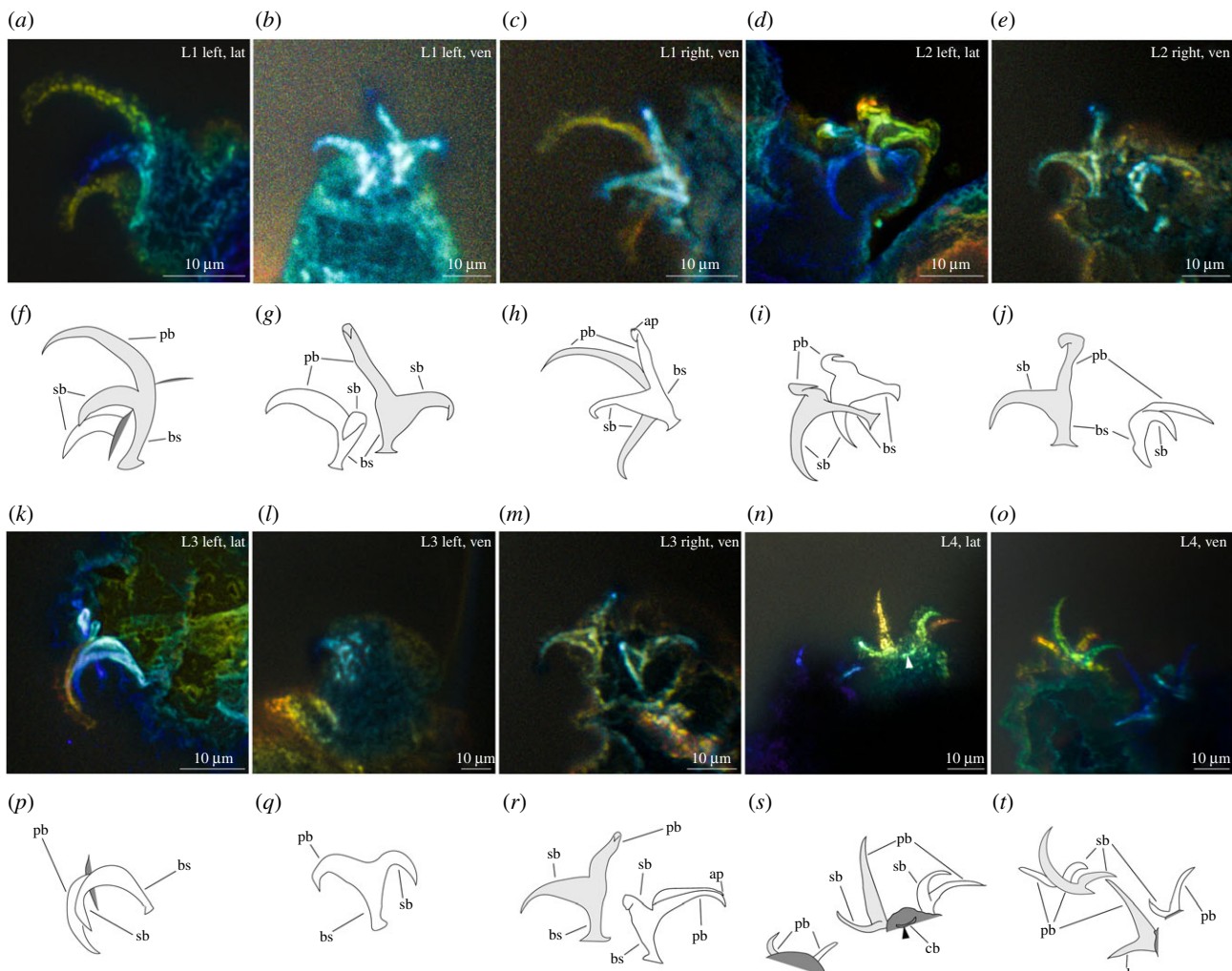

**Figure 3.** Claws of *Paradoryphoribius chronocaribbeus* gen. et sp. nov. from Miocene Dominican amber. (*a–e,k–o*) Structures photographed with autofluorescence under confocal microscope at 488 nm; different colours indicate *z*-depth, with violet to red gradient representing the shallowest to deepest planes, respectively; arrow indicates the cuticular bar. (*f–j,p–t*) Schematic drawing. Light grey shade—external (legs I–III) and posterior claws (leg IV); unshaded—internal (legs I–III) and anterior claws (leg IV); dark grey shade—leg portion; arrowhead—cuticular bar. ap, accessory point; bs, basal section; cb, cuticular bar; Ln, leg number; lat, lateral view; pb, primary branch; sb, secondary branch; ven, ventral view. (Online version in colour.)

*Etymology*: from the Greek 'chrono' (*khronos*)—meaning time—in reference to the age of the fossil taxon and '*caribbeus*' reflecting the region of the type locality.

*Type locality*: Dominican Republic mined from La Cumbre; amber from this region dates to the Miocene, with an approximate age of 16 Ma [28,29].

*Type material*: the initial amber specimen measured 2.8 × 2.1 × ∼0.5 cm and included other euarthropod synclusions, as well as a partial flower. Synclusions included an auchenorrhynchan hemipteran, a staphylinid beetle (Pselaphinae) and three ant workers belonging to distantly related lineages: *Neivamyrmex ectopus* (Dorylinae), *Nesomyrmex* sp. (Myrmicinae) and *Gnamptogenys* sp. (Ectatomminae). The tardigrade inclusion was trimmed out of the larger amber specimen with a water-fed diamond edge trim saw and polished with fine-grit emery papers on an EcoMet 30 (Buehler, Inc.) grinder polisher. The specimen is deposited in the American Museum of Natural History (AMNH) Division of Invertebrate of Zoology (AMNH DR-NJIT002).

*Diagnosis*: similar to the diagnosis of the genus.

*Description*: AMNH DR-NJIT002 consists of a complete, well-preserved individual that is most clearly observable from the left side in lateral and oblique views (figures 1 and 2; electronic supplementary material, figure 1*a,b*). The individual

has a total length (sagittal) of 559 μm. The body is surrounded by a rough-textured layer, but a portion of the cuticle is exposed and appears to be smooth (figure 1*b,c*, white region). A weakly outlined gibbosity-like structure can be observed on the dorsal side relative to leg III that appears to be ridge-like in shape, tapering on its edges and perpendicular to the body axis (figure 4*a,b*). This structure could also be a cuticular fold caused by the fossilization process. Owing to the rough layer covering, the presence of other dorsal cuticular structures and their exact number cannot be verified. In place of the left leg II is a large mass (figures 1 and 2), which probably corresponds to the body fluid that leaked out owing to compression during preservation. Eyespots were not observed.

AMNH DR-NJIT002 possesses *Isohypsibius*-type claws, with the claw pairs differing slightly in shape and size (figure 3; electronic supplementary material, figures S1*c–l* and S2). As shown in leg I, both claws have a wide basal section, but the external claw has a primary branch that appears to be substantially longer than the secondary branch (*br* = 57.24%; electronic supplementary material, table S1) (figure 3*b*). By contrast, the internal claw has a primary branch that appears to have a similar length as the secondary branch (*br* = 95.45%; electronic supplementary material, table S1) (figure 3*c*). It should be noted that the morphometric values of claws from

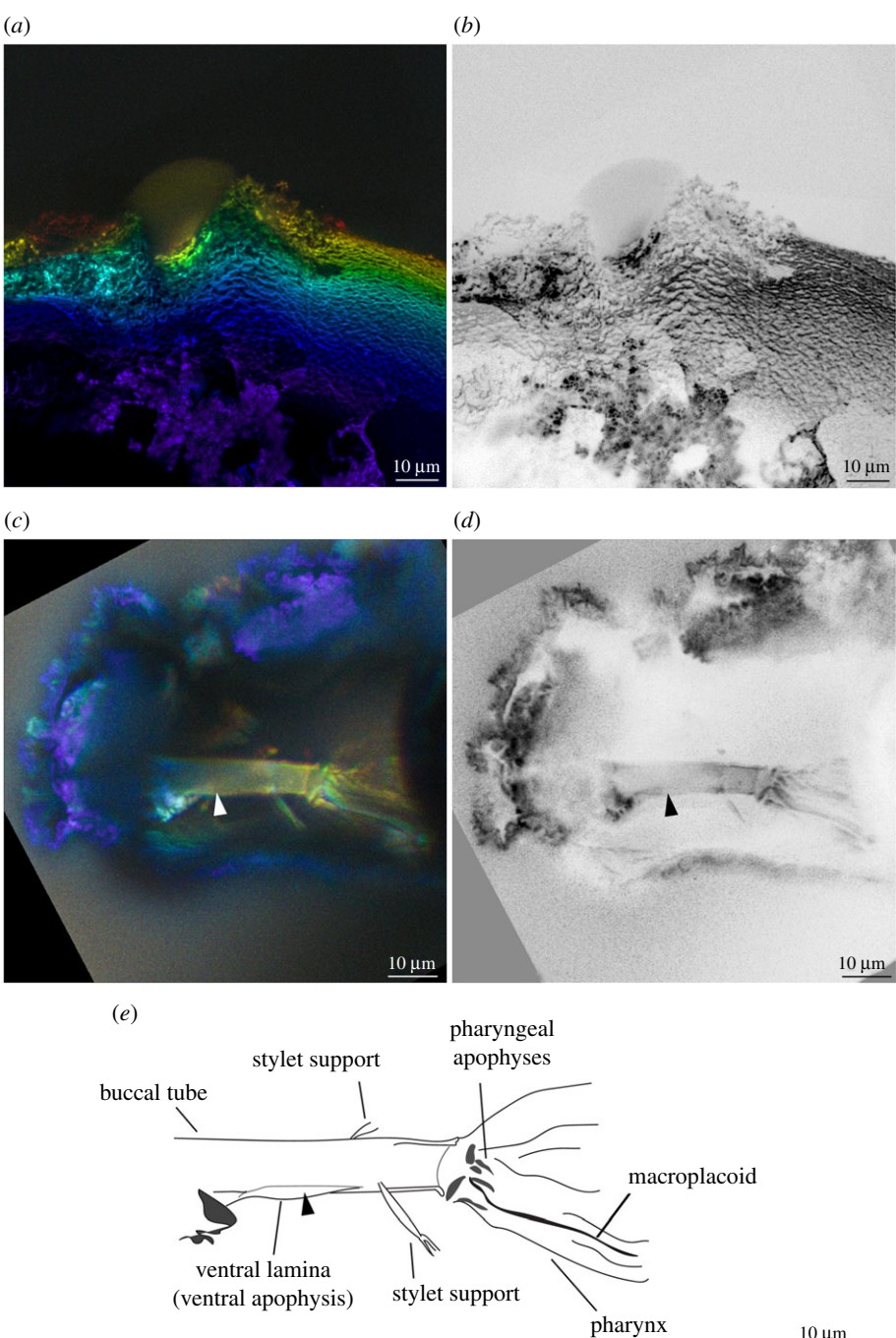

*(a)* *(b)*

*(c)* *(d)*

*(e)*

buccal tube

stylet support

pharyngeal apophyses

macroplacoid

ventral lamina (ventral apophysis)

stylet support

pharynx

10 µm

**Figure 4.** Left lateral view of the external and internal cuticular structures of *Paradoryphoribius chronocaribbeus* gen. et sp. nov. from Miocene Dominican amber. (*a*) Dorsal gibbosity-like structure photographed with autofluorescence under confocal microscope at 488 nm (different colours indicate *z*-depth, with violet to red gradient representing the shallowest to deepest planes, respectively). (*b*) Dorsal gibbosity-like structure viewed in inverted greyscale to highlight autofluorescence intensity (darker—more intense, lighter—least intense). (*c*) Buccal tube photographed with autofluorescence, viewed and presented similar to (*a*,*d*). Buccal tube viewed in inverted greyscale. (*e*) Schematic drawing. Arrowhead—ventral lamina. (Online version in colour.)

the other legs cannot be measured owing to their different planes of orientations and morphological distortions probably resulting from the taphonomic processes the sample underwent (e.g. primary branch of the external claw of the right leg II, figure 3*e*,*j*). Only one claw can be observed in the left leg III (figure 3*k*,*l*,*p*,*q*), but because two claws are observed in the right leg III (figure 3*m*,*r*), this absence is considered a taphonomic artefact. The position of this single claw on the leg suggests that it is probably the internal claw. In leg IV, the posterior and anterior claws have a similar size to the external and internal claws, respectively. Additionally, a cuticular bar can be seen in between the claws of the right leg IV (figure 3*n*,*s*; electronic supplementary material, figure S2 K, arrowhead). Accessory points were found in some of the claws, but

appear to be faint (figure 3*c*,*h*,*m*,*r*). Pseudolunules were not observed in the claws.

The anterior portion of the bucco-pharyngeal apparatus of AMNH DR-NJIT002 is covered by the rough-textured layer which did not allow the visualization of the mouth, peribuccal structures and oral cavity armature (figure 4*c*–*e*). Nevertheless, the posterior side showed that the sample has a rigid buccal tube possessing a narrow ventral lamina as the apophysis for the insertion of stylet muscles (figure 4*c*–*e*, arrowhead) extending almost to the insertion point of the stylet support. No dorsal apophysis is observed. Cuticular thickenings along the pharynx can be observed as pharyngeal apophyses and an undivided thin macroplacoid. Microplacoids are not observed and appear to be legitimately absent.

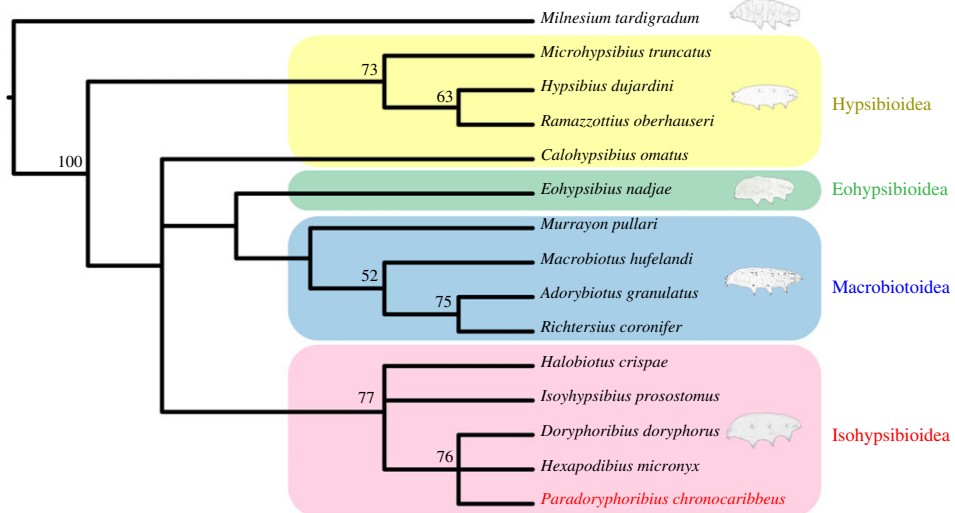

**Figure 5.** Phylogenetic result of the maximum-parsimony analysis using 28 morphological characters. Strict consensus tree of two most parsimonious trees obtained using implied weight setting ($k = 3$, consistency index = 0.62, retention index = 0.67). Node values indicate the support values above 50% obtained using symmetric resampling ($p = 15$) after 1000 replicates. Tardigrade figure for Eohypsibioidea modified from [27]. (Online version in colour.)

## (b) Results of phylogenetic analysis

*Paradoryphoribius chronocaribbeus* gen. et sp. nov. is consistently resolved within a monophyletic Isohypsibioidea under all treatments, supporting its systematic assignment in this extant superfamily. MP analysis using equal weighting obtained five most parsimonious trees (MPTs) (electronic supplementary material, file S3) while using implied weights ($k = 3$) obtained two MPTs (electronic supplementary material, file S4). The strict consensus tree of MPTs obtained using implied weights showed a clustering of *Pdo. chronocaribbeus* gen. et sp. nov. with all genera belonging to the superfamily Isohypsibioidea (figure 5; electronic supplementary material, figure S3). Specifically, it clustered with isohypsibioids (*Doryphoribius doryphorus* and *Hexapodibius micronyx*) which also possess ventral lamina as their ventral AISM. The same clustering was also observed in the strict consensus tree of MPTs obtained using equal weights (electronic supplementary material, figure S4). The result of the Bayesian analysis recovered a polytomous clade consisting of the fossil and the other isohypsibioids (electronic supplementary material, figure S5).

Aside from the isohypsibioids, all members of Hypsibioidea, except for *Calohypsibius ornatus*, clustered together in both MP and BI. Genera belonging to Macrobiotoidea clustered in all MP analyses, but not in BI. In these MP results, Eohypsibioidea was consistently recovered together with other macrobiotoids and showed a sister group relationship with Macrobiotoidea. This clustering relationship of Eohypsibioidea and Macrobiotoidea is also observed in recent molecular phylogenies of Eutardigrada [10,11]. Hypsibioidea was recovered as the sister group to other parachelans in MP under implied weights, but not in the other analyses. Thus, our phylogenetic analyses were not able to fully resolve the relationships between the parachelan superfamilies. This ambiguity is also observed in recent eutardigrade molecular phylogenies wherein even though Isohypsibioidea is sometimes recovered as the sister group to other parachelans, polytomies among the parachelan superfamilies are still observed depending on the genes, models and analyses used [10,11]. Lastly, the results of our morphological phylogeny are more similar to recent molecular phylogenies compared to the first morphological phylogeny that was done using all eutardigrade genera [26]. This is probably owing to the latter study having significantly more taxa than morphological characters used in their analyses.

## 4. Discussion

The discovery of *Pdo. chronocaribbeus* gen. et sp. nov. contributes towards a better understanding of the scant tardigrade fossil record, which contrasts with the long evolutionary history and cosmopolitan distribution of this lineage. Molecular-based divergence date estimates suggest an Ediacaran age for the most recent common ancestor of all extant Tardigrada [30,31] (figure 6). That such an ancient and widely distributed metazoan group has a palaeontological record comprising just three confidently ascribed fossils underscores the taphonomic challenges inherent in tardigrade preservation. While putative stem-group Orsten-tardigrade specimens have been recovered among a remarkable assemblage of Cambrian-age phosphatized microfossils, the remaining known fossil taxa are all preserved in amber dated to the Cretaceous or younger (figure 6a). Because of their minute size, largely non-biomineralized morphology and habitat preferences, tardigrades face a strong preservation bias as they would rarely encounter depositional conditions that would allow the preservation of diagnostic characters at this scale. Indeed, previously described amber tardigrade specimens ranged between 300 µm and 850 µm in length, while Orsten-tardigrade specimens range between 250 µm and 350 µm [2–4]. Amber fossilization can preserve submillimetric organisms with high fidelity, extreme examples include protists just 10–20 µm in length [33] and spirochete features less than a micron in size [34]. While there are other taphonomic windows that could theoretically allow for the preservation of tardigrade body fossils, for example, small carbonaceous fossils [35,36] in organic-rich marine shale, other factors such as decay or a lack of fine-scale sampling have contributed to an almost entirely amber-based fossil record so far.

The sparse tardigrade fossil record indicates that the preservation potential of these organisms, or at the very least the likelihood of fossil discovery, is almost entirely restricted to amber, which is also limited in its spatio-temporal occurrence. Although the oldest known fossil amber dates to the Carboniferous, *ca* 320 Ma [37], the oldest amber with euarthropod inclusions

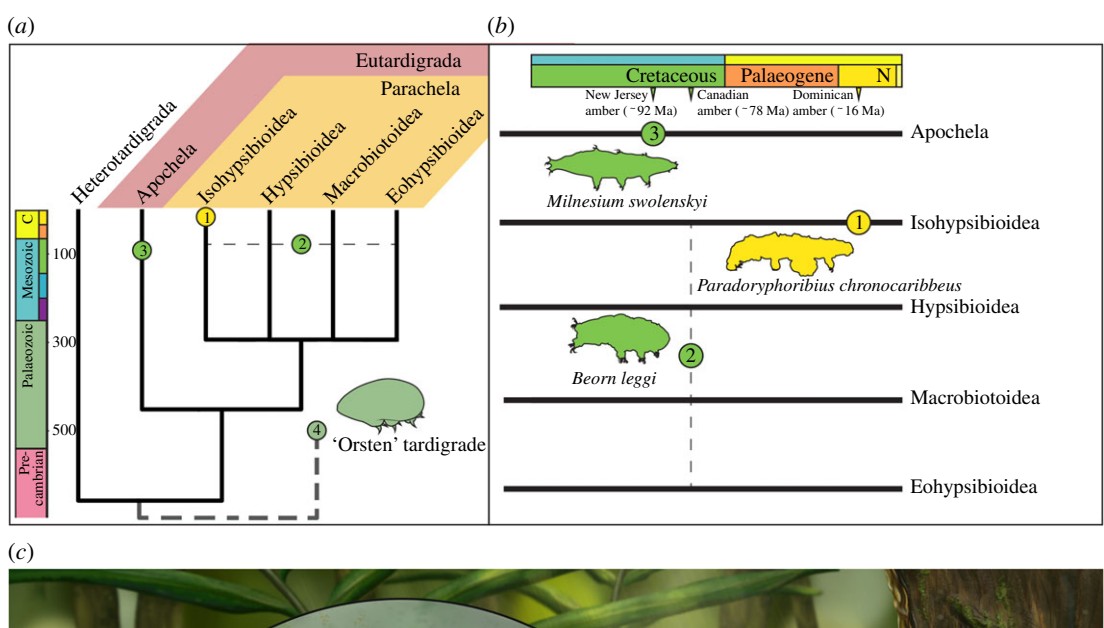

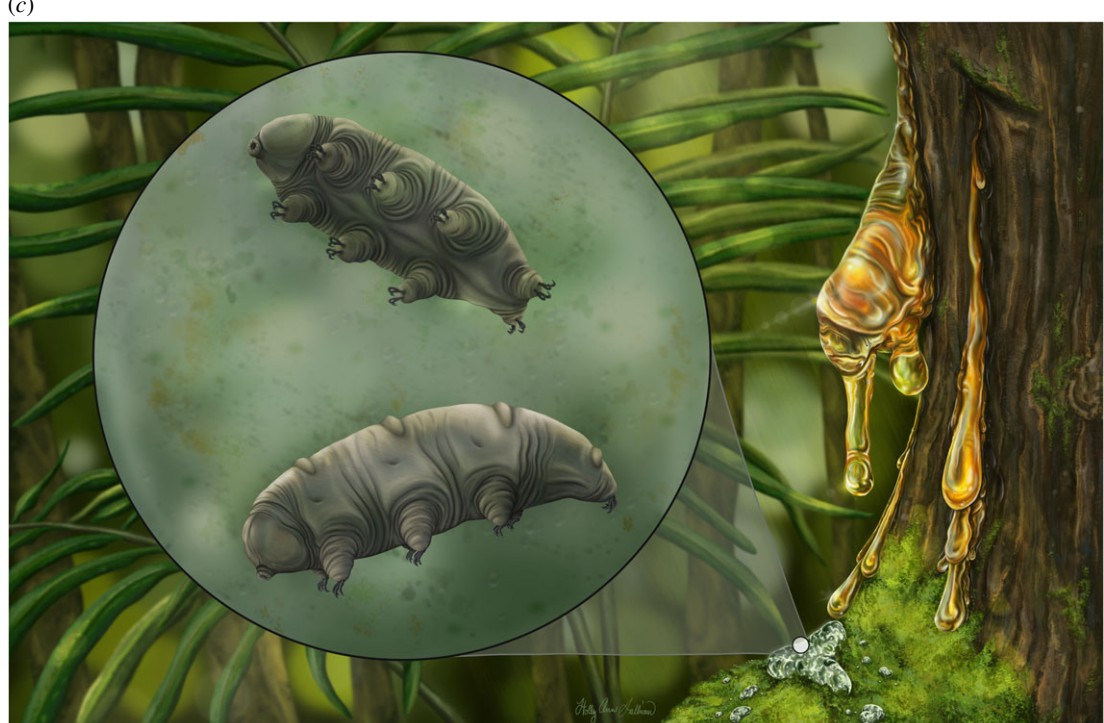

**Figure 6.** Phylogenetic and temporal summary of the tardigrade fossil record. (*a*) Simplified tardigrade phylogeny following topology from recent molecular analyses of [11] with taxonomic ranks *sensu* [32]; node ages correspond with mean molecular divergence estimates of Regier *et al.* [30]. The position of the undescribed 'Orsten' tardigrade as sister to all extant tardigrades reflects the putative stem-group hypotheses recently summarized by Guidetti & Bertolani [7]. (*b*) Eutardigrade fossil record. (1) *Paradoryphoribius chronocaribbeus* gen. et sp. nov. from Miocene age Dominican amber. (2) *Beorn leggi* from Campanian age Grassy Lake Canadian amber. Horizontal dotted line indicates uncertainty in the ordinal-level placement. (3) *Milnesium swolenskyi* from Turonian age Raritan amber from New Jersey. (4) 'Orsten' tardigrade from the middle Cambrian Kuonamka Formation in Siberia. (*c*) Artistic reconstruction of *Paradoryphoribius chronocaribbeus* gen. et sp. nov. Artist: Holly Sullivan. (Online version in colour.)

dates to the Late Triassic, *ca* 230 Ma, from the Dolomites in Italy [38]. Even then, only three taxa (a nematoceran fly and two eriophyoid mites), are known from the Dolomites from a screening of approximately 70 000 individual amber pieces [39] highlighting the rarity of this mode of preservation in the stratigraphically oldest deposits. Although there are scattered Jurassic amber deposits, none are known to contain significant numbers of diverse microinvertebrates. However, the production of fossil-bearing amber deposits increased markedly with the Cretaceous Terrestrial Revolution and expansion of angiosperms [40].

Most fossil amber, and thus most sampling opportunity for tardigrade inclusions, dates to the Late Cretaceous and Cenozoic. The most heavily sampled and studied are Burmese amber (Cretaceous), Baltic amber (Eocene) and Dominican amber (Miocene). More than 3000 euarthropod species have been described from Baltic amber [29], nearly 1500 total species in Burmese amber [41] and nearly 1400 species from Dominican amber [42]. It is notable that previously described fossil tardigrade taxa are known from the less intensively sampled Canadian and New Jersey amber, with just over 100 and approximately 130 species reported, respectively [43,44]. A possible explanation may be the level of attention to detail involved in the amber screening process, given that Burmese, Baltic and Dominican amber are primarily commercially mined and prepared, and thus introduce a substantial collector bias for large and visually appealing inclusions [45]. This highlights the need for precautions in preparing and analysing samples of ambers since they might contain valuable tardigrade inclusions.

*Paradoryphoribius chronocaribbeus* gen. et sp. nov. is, to our knowledge, the only fossil tardigrade described from the

Cenozoic to date, as well as the only definitive member of the superfamily Isohypsibioidea, which has been sometimes recovered as a sister group to all other Parachela [10,11,46]. The new taxon, therefore, provides a minimum age for the superfamily. Whether *Pdo. chronocaribbeus* gen. et sp. nov. is a member of a wholly extinct tardigrade lineage or perhaps has affinities to undiscovered living taxa is not yet clear.

Because of the relatively young age of the Dominican amber, its euarthropod fauna is largely modern and most species are attributable to extant lineages among well-sampled groups. For example, although ants comprise between 25 and 35% of all Dominican amber inclusions and there are currently more than 80 described species from this deposit [47], only two of them are placed within extinct genera [42]. Some extant taxa were first described in Dominican amber prior to their discovery in extant neotropical communities, such as the ant genera *Gracilidris* and *Leptomyrmex* [48,49]. These 'Lazarus taxa' were initially known only from Dominican amber material until their discovery in South America decades later. Caribbean biogeography is complex with several proposed hypotheses for dispersal among the Greater and Lesser Antilles. Additional Caribbean surveys will improve understanding of the systematic position of *Pdo. chronocaribbeus* gen. et sp. nov. and may contribute to a clearer picture of the dispersal pathways of tardigrades across this region. Of particular concern to Hispaniolan biogeography is a proposed interconnected landmass spanning the Greater Antilles and Aves Ridge called GAARlandia [28]. Importantly, GAARlandia would have connected the Greater Antilles to mainland South America up until the Oligocene. The broad limno-terrestrial tardigrade community of Central America harbours a significant number of endemics and is distinct from North and South American fauna [50]. A similar pattern exists among Dominican amber and Mesoamerican termites [51]. However, South American affinities have been noted for Dominican amber spiders [52], and there is significant heterogeneity among inferred dispersal histories across Caribbean euarthropods [53].

# 5. Conclusion

The discovery of *Pdo. chronocaribbeus* gen. et sp. nov. underscores the rarity of tardigrade fossils and their preferential preservation potential in amber. Confocal fluorescence microscopy shows that taxonomically important characters, particularly the claws and bucco-pharyngeal apparatus, are preserved in the single specimen available. This imaging technique has also yielded similar results for *Mil. swolenskyi* (fig. 5.1 in [7]), emphasizing its use for studying the morphology of microinvertebrates preserved in amber, including that of the still problematic *Beo. leggi* [2]. Given the rarity of fossil tardigrades, the systematic placements of *Beo. leggi*, *Mil. swolenskyi* and *Pdo. chronocaribbeus* gen. et sp. nov. are critical for calibrating molecular estimates for the deep origin of this phylum and provide new insights on important major evolutionary events, such as the miniaturization of their body plan and the terrestrialization of eutardigrades [54,55].

Ethics. Dominican amber is commercially mined and marketed, the only restriction is on the sale of unprocessed amber—it is illegal to remove any amber from the country without local artisans first shaping and polishing the material. This specimen was purchased from one such commercial artisan source: Jorge Martinez, a licensed and long-time dealer in Dominican amber.

Data accessibility. All data used and generated are provided in either the main text or in the electronic supplementary material.

Authors' contributions. M.A.M.: conceptualization, data curation, formal analysis, methodology, project administration, writing—original draft, writing—review and editing; N.R.: data curation, methodology, resources, writing—review and editing; BE.B.: data curation, methodology, resources, writing—review and editing; J.O.: conceptualization, data curation, formal analysis, funding acquisition, methodology, project administration, resources, supervision, writing—review and editing; P.B.: conceptualization, data curation, formal analysis, funding acquisition, methodology, project administration, resources, supervision, writing—original draft, writing—review and editing. All authors gave final approval for publication and agreed to be held accountable for the work performed therein.

Competing interests. We declare we have no competing interests.

Funding. Published by a grant from the Wetmore Colles Fund. N.R. was funded by the 2018–2019 Fulbright Non-US Visiting Scholar Program of the French-American Fulbright Commission. B.E.B. is funded by an Alexander von Humboldt Stiftung Research Fellowship.

Acknowledgements. We thank Jorge Martinez of Gurabo, Santiago for providing the type specimen and the Harvard Center for Biological Imaging (HCBI) for infrastructure and support in microscope imaging. The authors also thank the editor and anonymous reviewers for providing helpful comments that improved the manuscript.

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
