## [Peer Review File · Proceedings of the Royal Society B: Biological Sciences]

Review History

RSPB-2021-0659.R0 (Original submission)

Review form: Reviewer 1

Recommendation

Major revision is needed (please make suggestions in comments)

Scientific importance: Is the manuscript an original and important contribution to its field?

Excellent

General interest: Is the paper of sufficient general interest?

Excellent

Quality of the paper: Is the overall quality of the paper suitable?

Marginal

Is the length of the paper justified?

No

Should the paper be seen by a specialist statistical reviewer?

No

Do you have any concerns about statistical analyses in this paper? If so, please specify them explicitly in your report.

Yes

It is a condition of publication that authors make their supporting data, code and materials available - either as supplementary material or hosted in an external repository. Please rate, if applicable, the supporting data on the following criteria.

Is it accessible?

Yes

Is it clear?

Yes

Is it adequate?

No

Do you have any ethical concerns with this paper?

No

Comments to the Author

The discovery of a fossil tardigrade in amber is highly significant, extremely rare, and should be published. Sadly I must say this paper, although extremely well written with the appropriate references, must be substantially revised because the identification of the genus is incorrect. Based on the figures presented, the specimen can only be identified to superfamily Isohypsibioidae. There is insufficient information to determine whether it is family Isohypsibiidae (Ursulinius) or family Doryphoribiidae (Doryphoribius), which are primarily separated by the absence (Ursulinius) or presence (Doryphoribius) of the ventral lamina on the buccal tube. Unfortunately the buccal-pharyngeal apparatus, is not visible in the specimen. Based on claws, the other essential characters (accessory points, pseudolunules, cuticular bars) are not clearly discernable with confocal microscopy. I encourage the authors to revise and resubmit after carefully evaluating my extensive comments in the manuscript. The discovery of a new fossil tardigrade is highly significant and exciting and provides important data on the evolution of tardigrades.

Review form: Reviewer 2

Recommendation

Accept with minor revision (please list in comments)

Scientific importance: Is the manuscript an original and important contribution to its field?

Acceptable

General interest: Is the paper of sufficient general interest?

Acceptable

Quality of the paper: Is the overall quality of the paper suitable?

Good

Is the length of the paper justified?

Yes

Should the paper be seen by a specialist statistical reviewer?

No

Do you have any concerns about statistical analyses in this paper? If so, please specify them explicitly in your report.

No

It is a condition of publication that authors make their supporting data, code and materials available - either as supplementary material or hosted in an external repository. Please rate, if applicable, the supporting data on the following criteria.

Is it accessible?

Yes

Is it clear?

Yes

Is it adequate?

Yes

Do you have any ethical concerns with this paper?

No

Comments to the Author

I leave the decision on whether the paper is of Proceedings impact to the editors. The taphonomic and temporal significance of the find are noted in the abstract but both are quite minor. We go from two occurrences of tardigrades in amber to three, a number so small that a 50% increase is not very compelling. The Miocene age of the fossil has little impact on the tardigrade time tree because the divergence of Isohypsibioidea within Eutardigrada was already constrained by any of the plausible placements of the Cretaceous Beorn.

On p. 7, line 18, indicate the geographic distributions of *Dianeia* and *Ursulinius* to allow the reader to better evaluate the likelihood of having been present in the Caribbean in the Miocene but having gone locally extinct.

The generic assignment draws partly on the presence of dorsal gibbosities. I have to say that these are not tremendously obvious, barely being distinguishable from other areas of the cuticle. Compared to SEM images of gibbosities in extant *Doryphoribius*, I can almost be convinced by the dorsal one on segment 3 in Fig. 1B but I can't really see anything in Figs. 1A and B that has an outline corresponding to the larger, more lateral "gibbosity" drawn in Fig. 1C.

Trivial edits:

P. 3, line 13: indicates that Cedar Lake is in Manitoba.

P. 3, line 16: "Its precise affinities remain..."

P. 7, line 28: "...dates to the Miocene..."

References 30 and 33: check that "Miocene" and "Carboniferous" are really not capitalised in the titles.

Review form: Reviewer 3

Recommendation

Major revision is needed (please make suggestions in comments)

Scientific importance: Is the manuscript an original and important contribution to its field?

Excellent

General interest: Is the paper of sufficient general interest?

Good

Quality of the paper: Is the overall quality of the paper suitable?

Good

Is the length of the paper justified?

Yes

Should the paper be seen by a specialist statistical reviewer?

No

Do you have any concerns about statistical analyses in this paper? If so, please specify them explicitly in your report.

No

It is a condition of publication that authors make their supporting data, code and materials available - either as supplementary material or hosted in an external repository. Please rate, if applicable, the supporting data on the following criteria.

Is it accessible?

Yes

Is it clear?

Yes

Is it adequate?

Yes

Do you have any ethical concerns with this paper?

No

Comments to the Author

Dear Authors,

Your manuscript is well-written and your Dominican amber specimen represents an important contribution to the very sparse tardigrade fossil record. The manuscript can, as such, be viewed as excellent. I, however, have major concerns with your assignment of the fossil. You have no data on essential buccal-pharyngeal structures, such as peribuccal sensory structures, pharyngeal placoids, apophyses for the stylet muscle and whether or not a ventral lamina is present on the buccal tube. Based on the limited morphological data that you have been able to extract from the fossil (i.e. claw morphology and possible presence of dorsal gibbosities), I would suggest that you erect a new genus and place the specimen as *incertae sedis* within Isohypsibioidea. I am therefore recommending a major revision. Below follows specific comments and suggestions to various paragraphs of your manuscript.

Page 4, lines 5-8. The introduction would benefit from being broadened a bit in order to better attract general interest. For the broader audience I would expect at least 4-5 references to recent

reviews or overviews on tardigrade morphology, phylogeny and taxonomy, extreme stress tolerance and survival strategies.

Page 5, Lines 11-13. I do not agree that the limited morphological data presented in this manuscript, i.e. selected claw branch length ratios and a possible presence of dorsal gibbosities (these are not clearly visible from the photos presented) can support the suggested assignment of the fossil to *Doryphoribius*. You simply lack the necessary data on essential buccal-pharyngeal structures in order to do this. The claim that the preserved morphology allows you to erect a new *Doryphoribius* species and assign the fossil to Doryphoribiidae represents an over-interpretation of the data at hand.

Page 5, Line 24-25. In what sense is Tardigrada a "cryptic metazoan phylum"? Please explain or reformulate.

Page 6, Line 20-21: I see 13 (not 10) species listed in your "Supplementary Sheet 2".

Page 6, Line 21-23: I suggest that you avoid using statistical terms, such as "mean", when you refer to data from your fossil, where you only have a single data point (n=1).

Page 7, Line 17: Eutardigrada was erected by Richters in 1926. Please provide the correct authority! The text should read: Class Eutardigrada Richters, 1926

Page 7, Line 26: Yes, the ventral lamina is a defining character of genus *Doryphoribius*, but you don't know whether your specimen has a ventral lamina or not. Accordingly your data do not provide any support that your specimen actually belongs to this genus.

Page 8, Lines 1-21: The "Remarks" section should be re-written, clearly acknowledging the fact that you lack defining data on buccal-pharyngeal structures, including (but not limited to) evidence of a ventral lamina on the buccal tube. My advice would be to avoid any over-interpretation of the limited morphological data that you were able to extract from the specimen. I suggest that you put focus on the data at hand, i.e. claw morphology. Along this line, I think that it would be very informative for the reader if you provide a more comprehensive account on isohypsibiid claw structure, in the form of schematic drawings and at the same time provided drawings of the claws on each leg of the fossil specimen.

Page 9, Lines 1-5: In case data is published on any of these synclusions, please provide references.

Page 9, Line 8: move abbreviation (AMNH): American Museum of Natural History (AMNH), Division of Invertebrate of Zoology.

Page 9, Line 10: Regarding the "weakly outlined dorsal gibbosities". I am not able to recognize these in Figure 1B. If possible, please provide a magnification of this area of the fossil.

Page 9, Line 11: "differing" instead of "differ".

Page 9 & 10, Lines 27-31 & 1-16. At present your description of the claws is rather hard to follow, and I would guess almost impossible to understand for a person that is not familiar with tardigrade claw morphology. Please help the reader and make sure that all details, such as "wide common tract", "triangular claw base", "external", "internal", "posterior" and "anterior" claws, "primary" and "secondary" branches etc. are clearly indicated on Figures 3A-H (e.g. with arrows and/or abbreviations). Please also see my suggestion above regarding schematic drawings of the claws on each leg – I think this would really improve the readability of your manuscript.

Page 10, Line 20: Regarding "claw structure (claw pairs slightly different from each other and each claw has br <70%)". According to Table 1 the claim that "br <70%" does not apply to all claw

pairs. Please explain.

Page 10, Lines 21-26: this represents an over-interpretation of your data.

Pages 10, 11 & 12, Lines 28-30, 1-31 & 1-11. I do not agree that the limited morphological data available from this fossil can support the suggested assignment to *Doryphoribius* (see my comments above). Accordingly, I suggest that you leave out the comparison to extant *Doryphoribius* species.

Page 14, Lines 1-5: As I do not agree with the suggested assignment to *Doryphoribius*, I would also suggest that you delete or reformulate these two sentences.

Figure 1: Please provide a magnification of the area with gibbosities.

Figure 3: Insert labels to mark details, such as "common tract", "triangular base", "external" and "internal" claws, "primary" and "secondary" branches etc. Also please provide drawings of the claws on each leg.

Figure 5: Consider also inserting the heterotardigrade fossil that was found together with *Beorn leggi* on this figure.

Table 1. Given the difficulties in obtaining accurate measurement, I would leave out decimals in the calculated br values. To ease readability please explain "br" in table text.

Supplementary Figure 1: suggest you avoid using the term "mean", when you refer to the length of your fossil.

Supplementary Figure 2: delete the repetition "under confocal microscope with autofluorescence at 488 nm" (Lines 3-4).

Decision letter (RSPB-2021-0659.R0)

07-May-2021

Dear Mr Mapalo:

I am writing to inform you that your manuscript RSPB-2021-0659 entitled "A tardigrade in Dominican amber" has, in its current form, been rejected for publication in Proceedings B.

This action has been taken on the advice of referees, who have recommended that substantial revisions are necessary. With this in mind we would be happy to consider a resubmission, provided the comments of the referees are fully addressed. However please note that this is not a provisional acceptance.

If you resubmit, please do provide a copy of a receipt of this amber purchase that includes contact information of Mr. Martinez and his business, for our records.

Sincerely,
 Dr John Hutchinson, Editor
<mailto:proceedingsb@royalsociety.org>

Associate Editor

Comments to Author:

Dear Authors

You will see that the reviewers have given the manuscript due attention and note that it is well presented and another fossil tardigrade is of importance to be brought into the literature.

However, all are concerned about the specificity of your diagnosis and whether the characters supporting this assignment actually can be determined and advise to leave the fossil in a higher ranking nomenclature.

Unless you can substantiate your claims with better illustrations/evidence you need to decide if you can compel the reviewers or take their advice to heart and revise the manuscript to assign the specimen with a more cautious systematic treatment.

Reviewer(s)' Comments to Author:

Referee: 1

Comments to the Author(s)

The discovery of a fossil tardigrade in amber is highly significant, extremely rare, and should be published. Sadly I must say this paper, although extremely well written with the appropriate references, must be substantially revised because the identification of the genus is incorrect. Based on the figures presented, the specimen can only be identified to superfamily Isohypsibioidae. There is insufficient information to determine whether it is family Isohypsibiidae (Ursulinius) or family Doryphoribiidae (Doryphoribius), which are primarily separated by the absence (Ursulinius) or presence (Doryphoribius) of the ventral lamina on the buccal tube. Unfortunately the buccal-pharyngeal apparatus, is not visible in the specimen. Based on claws, the other essential characters (accessory points, pseudolunules, cuticular bars) are not clearly discernable with confocal microscopy. I encourage the authors to revise and resubmit after carefully evaluating my extensive comments in the manuscript. The discovery of a new fossil tardigrade is highly significant and exciting and provides important data on the evolution of tardigrades.

Referee: 2

Comments to the Author(s)

I leave the decision on whether the paper is of Proceedings impact to the editors. The taphonomic and temporal significance of the find are noted in the abstract but both are quite minor. We go from two occurrences of tardigrades in amber to three, a number so small that a 50% increase is not very compelling. The Miocene age of the fossil has little impact on the tardigrade time tree because the divergence of Isohypsibioidea within Eutardigrada was already constrained by any of the plausible placements of the Cretaceous Beorn.

On p. 7, line 18, indicate the geographic distributions of *Dianeia* and *Ursulinius* to allow the reader to better evaluate the likelihood of having been present in the Caribbean in the Miocene but having gone locally extinct.

The generic assignment draws partly on the presence of dorsal gibbosities. I have to say that these are not tremendously obvious, barely being distinguishable from other areas of the cuticle. Compared to SEM images of gibbosities in extant *Doryphoribius*, I can almost be convinced by the dorsal one on segment 3 in Fig. 1B but I can't really see anything in Figs. 1A and B that has an outline corresponding to the larger, more lateral "gibbosity" drawn in Fig. 1C.

Trivial edits:

P. 3, line 13: indicates that Cedar Lake is in Manitoba.

P. 3, line 16: "Its precise affinities remain..."

P. 7, line 28: "...dates to the Miocene..."

References 30 and 33: check that "Miocene" and "Carboniferous" are really not capitalised in the titles.

Referee: 3

Comments to the Author(s)

Dear Authors,

Your manuscript is well-written and your Dominican amber specimen represents an important contribution to the very sparse tardigrade fossil record. The manuscript can, as such, be viewed as excellent. I, however, have major concerns with your assignment of the fossil. You have no data on essential buccal-pharyngeal structures, such as peribuccal sensory structures, pharyngeal placoids, apophyses for the stylet muscle and whether or not a ventral lamina is present on the buccal tube. Based on the limited morphological data that you have been able to extract from the fossil (i.e. claw morphology and possible presence of dorsal gibbosities), I would suggest that you erect a new genus and place the specimen as *incertae sedis* within Isohypsibioidea. I am therefore recommending a major revision. Below follows specific comments and suggestions to various paragraphs of your manuscript.

Page 4, lines 5-8. The introduction would benefit from being broadened a bit in order to better attract general interest. For the broader audience I would expect at least 4-5 references to recent reviews or overviews on tardigrade morphology, phylogeny and taxonomy, extreme stress tolerance and survival strategies.

Page 5, Lines 11-13. I do not agree that the limited morphological data presented in this manuscript, i.e. selected claw branch length ratios and a possible presence of dorsal gibbosities (these are not clearly visible from the photos presented) can support the suggested assignment of the fossil to *Doryphoribius*. You simply lack the necessary data on essential buccal-pharyngeal structures in order to do this. The claim that the preserved morphology allows you to erect a new

Doryphoribius species and assign the fossil to Doryphoribiidae represents an over-interpretation of the data at hand.

Page 5, Line 24-25. In what sense is Tardigrada a "cryptic metazoan phylum"? Please explain or reformulate.

Page 6, Line 20-21: I see 13 (not 10) species listed in your "Supplementary Sheet 2".

Page 6, Line 21-23: I suggest that you avoid using statistical terms, such as "mean", when you refer to data from your fossil, where you only have a single data point (n=1).

Page 7, Line 17: Eutardigrada was erected by Richters in 1926. Please provide the correct authority! The text should read: Class Eutardigrada Richters, 1926

Page 7, Line 26: Yes, the ventral lamina is a defining character of genus *Doryphoribius*, but you don't know whether your specimen has a ventral lamina or not. Accordingly your data do not provide any support that your specimen actually belongs to this genus.

Page 8, Lines 1-21: The "Remarks" section should be re-written, clearly acknowledging the fact that you lack defining data on buccal-pharyngeal structures, including (but not limited to) evidence of a ventral lamina on the buccal tube. My advice would be to avoid any over-interpretation of the limited morphological data that you were able to extract from the specimen. I suggest that you put focus on the data at hand, i.e. claw morphology. Along this line, I think that it would be very informative for the reader if you provide a more comprehensive account on isohypsibiid claw structure, in the form of schematic drawings and at the same time provided drawings of the claws on each leg of the fossil specimen.

Page 9, Lines 1-5: In case data is published on any of these synclusions, please provide references.

Page 9, Line 8: move abbreviation (AMNH): American Museum of Natural History (AMNH), Division of Invertebrate of Zoology.

Page 9, Line 10: Regarding the "weakly outlined dorsal gibbositities". I am not able to recognize these in Figure 1B. If possible, please provide a magnification of this area of the fossil.

Page 9, Line 11: "differing" instead of "differ".

Page 9 & 10, Lines 27-31 & 1-16. At present your description of the claws is rather hard to follow, and I would guess almost impossible to understand for a person that is not familiar with tardigrade claw morphology. Please help the reader and make sure that all details, such as "wide common tract", "triangular claw base", "external", "internal", "posterior" and "anterior" claws, "primary" and "secondary" branches etc. are clearly indicated on Figures 3A-H (e.g. with arrows and/or abbreviations). Please also see my suggestion above regarding schematic drawings of the claws on each leg - I think this would really improve the readability of your manuscript.

Page 10, Line 20: Regarding "claw structure (claw pairs slightly different from each other and each claw has br <70%)". According to Table 1 the claim that "br <70%" does not apply to all claw pairs. Please explain.

Page 10, Lines 21-26: this represents an over-interpretation of your data.

Pages 10, 11 & 12, Lines 28-30, 1-31 & 1-11. I do not agree that the limited morphological data available from this fossil can support the suggested assignment to *Doryphoribius* (see my comments above). Accordingly, I suggest that you leave out the comparison to extant *Doryphoribius* species.

Page 14, Lines 1-5: As I do not agree with the suggested assignment to *Doryphoribius*, I would also suggest that you delete or reformulate these two sentences.

Figure 1: Please provide a magnification of the area with gibbosities.

Figure 3: Insert labels to mark details, such as "common tract", "triangular base", "external" and "internal" claws, "primary" and "secondary" branches etc. Also please provide drawings of the claws on each leg.

Figure 5: Consider also inserting the heterotardigrade fossil that was found together with *Beorn leggi* on this figure.

Table 1. Given the difficulties in obtaining accurate measurement, I would leave out decimals in the calculated br values. To ease readability please explain "br" in table text.

Supplementary Figure 1: suggest you avoid using the term "mean", when you refer to the length of your fossil.

Supplementary Figure 2: delete the repetition "under confocal microscope with autofluorescence at 488 nm" (Lines 3-4).

Author's Response to Decision Letter for (RSPB-2021-0659.R0)

See Appendix A.

RSPB-2021-1760.R0

Review form: Reviewer 1

Recommendation

Accept with minor revision (please list in comments)

Scientific importance: Is the manuscript an original and important contribution to its field?

Excellent

General interest: Is the paper of sufficient general interest?

Excellent

Quality of the paper: Is the overall quality of the paper suitable?

Excellent

Is the length of the paper justified?

Yes

Should the paper be seen by a specialist statistical reviewer?

No

Do you have any concerns about statistical analyses in this paper? If so, please specify them explicitly in your report.

No

It is a condition of publication that authors make their supporting data, code and materials available - either as supplementary material or hosted in an external repository. Please rate, if applicable, the supporting data on the following criteria.

Is it accessible?

Yes

Is it clear?

Yes

Is it adequate?

Yes

Do you have any ethical concerns with this paper?

No

Comments to the Author

Thank you for making the changes requested in the previous review. This is an intriguing paper that describes a new genus and new species of a tardigrade in Dominican amber, adding to our scant knowledge of fossil tardigrades. I found a few typographical/spelling errors that can be easily corrected. These are listed below:

1. Page 8, Line 25: correct spelling of Hypsibioidea: Hypsibiidae
2. Page 13, Line 17: I think you meant to have a period [.] to end the sentence at "later." Then start a new sentence with "Caribbean...."
3. Page 16, Reference #32: italicize the genus & species of the termite; capitalize Miocene.
4. Page 16, Reference #33: capitalize Paleozoic
5. Page 17, Reference #43: Cambrian Spence Shale Lagerstätte should be capitalized.
6. Page 17, reference #44: There should be a dash rather than a hyphen between tardigrades--adding genes
7. Page 17, Reference: italicize Leptomyrmex
8. Figs 1-4, and Supplementary figures: add comma between "planes" and "respectively"
9. Fig 4 and Suppl Fig 1 and Suppl Fig 2: use a dash instead of a hyphen between darker - more intense and between lighter - less intense
10. Suppl Figs. 3, 4, and 5: correct spelling of Eohypsibioidea

Review form: Reviewer 2

Recommendation

Accept with minor revision (please list in comments)

Scientific importance: Is the manuscript an original and important contribution to its field?

Good

General interest: Is the paper of sufficient general interest?

Good

Quality of the paper: Is the overall quality of the paper suitable?

Excellent

Is the length of the paper justified?

Yes

Should the paper be seen by a specialist statistical reviewer?

No

Do you have any concerns about statistical analyses in this paper? If so, please specify them explicitly in your report.

No

It is a condition of publication that authors make their supporting data, code and materials available - either as supplementary material or hosted in an external repository. Please rate, if applicable, the supporting data on the following criteria.

Is it accessible?

Yes

Is it clear?

Yes

Is it adequate?

Yes

Do you have any ethical concerns with this paper?

No

Comments to the Author

This paper is much improved in its revised version. New images of the specimens, schematics of the claws and buccal tube, and a morphological phylogenetic analysis to justify the classification of the fossil are all substantial improvements. The dorsal gibbosity that underwhelmed me last time is now convincing. I am happy to sign off on this version and leave the tardigrade specialist reviewers to check the revision.

Just one suggestion. I don't see the point of Supplementary Figure 3. Figure 5 shows the G/C values for the reasonably supported nodes in this implied weighted tree. All Supplementary Figure 5 adds is a few nodes that have G/C values of 3, 4, 6 and 47. These poorly supported nodes really don't warrant reporting.

Review form: Reviewer 4

Recommendation

Accept with minor revision (please list in comments)

Scientific importance: Is the manuscript an original and important contribution to its field?

Good

General interest: Is the paper of sufficient general interest?

Good

Quality of the paper: Is the overall quality of the paper suitable?

Excellent

Is the length of the paper justified?

Yes

Should the paper be seen by a specialist statistical reviewer?

No

Do you have any concerns about statistical analyses in this paper? If so, please specify them explicitly in your report.

No

It is a condition of publication that authors make their supporting data, code and materials available - either as supplementary material or hosted in an external repository. Please rate, if applicable, the supporting data on the following criteria.

Is it accessible?

Yes

Is it clear?

Yes

Is it adequate?

Yes

Do you have any ethical concerns with this paper?

No

Comments to the Author

The paper is interesting and well written.

My only concern is with the presence of gibbosities in the animals. The authors do not see them, they saw only a fold that they interpreted as a gibbosity. If the authors look at the bibliography there are examples in which animals present folds in the cuticle similar to those that they found (e.g. <https://doi.org/10.1080/14772000.2020.1771469>, <https://doi.org/10.1038/s41598-021-83627-w>), but that do not represent gibbosities, they are just folds due to specimen preparation. The authors need to be more cautious in writing that gibbosities are present. They should write that gibbosities may be present (given the data they present), but that the highlighted gibbosity could be the result of a cuticle fold due to fossilization and/or cuticle breakage at the most anterior part of the dorsal fold (which they interpret as gibbosity). Therefore, in the definition of the taxon, they have to write that the gibbosities could be present, and not that they are certainly present.

Decision letter (RSPB-2021-1760.R0)

02-Sep-2021

Dear Mr Mapalo:

Your manuscript has now been peer reviewed and the reviews have been assessed by an Associate Editor. The reviewers' comments (not including confidential comments to the Editor) and the comments from the Associate Editor are included at the end of this email for your reference. As you will see, the reviewers and the Editors have raised some concerns with your manuscript and we would like to invite you to revise your manuscript to address them.

Research ethics:

Use of animals and field studies:

It is a condition of publication that you make available the data and research materials supporting the results in the article (<https://royalsociety.org/journals/authors/author-guidelines/#data>). Datasets should be deposited in an appropriate publicly available repository and details of the associated accession number, link or DOI to the datasets must be included in the Data Accessibility section of the article (<https://royalsociety.org/journals/ethics-policies/data-sharing-mining/>). Reference(s) to datasets should also be included in the reference list of the article with DOIs (where available).

Please submit a copy of your revised paper within three weeks. If we do not hear from you within this time your manuscript will be rejected. If you are unable to meet this deadline please let us know as soon as possible, as we may be able to grant a short extension.

Best wishes,
Dr John Hutchinson, Editor
mailto:proceedingsb@royalsociety.org

Associate Editor
Comments to Author:

You will see that the referees are happy with the manuscript and only suggest a few issues to consider, such as some corrections/typos (reviewer 3), the nature of gibbosities or not (reviewer 2) and which nodes to report on in the supplement if poorly supported (reviewer 1).

This should all be manageable revisions.

Yours Jakob

Reviewer(s)' Comments to Author:

Referee: 2

Comments to the Author(s).

This paper is much improved in its revised version. New images of the specimens, schematics of the claws and buccal tube, and a morphological phylogenetic analysis to justify the classification of the fossil are all substantial improvements. The dorsal gibbosity that underwhelmed me last time is now convincing. I am happy to sign off on this version and leave the tardigrade specialist reviewers to check the revision.

Just one suggestion. I don't see the point of Supplementary Figure 3. Figure 5 shows the G/C values for the reasonably supported nodes in this implied weighted tree. All Supplementary Figure 5 adds is a few nodes that have G/C values of 3, 4, 6 and 47. These poorly supported nodes really don't warrant reporting.

Referee: 4

Comments to the Author(s).

The paper is interesting and well written.

My only concern is with the presence of gibbosities in the animals. The authors do not see them, they saw only a fold that they interpreted as a gibbosity. If the authors look at the bibliography

there are examples in which animals present folds in the cuticle similar to those that they found (e.g. <https://doi.org/10.1080/14772000.2020.1771469>, <https://doi.org/10.1038/s41598-021-83627-w>), but that do not represent gibbositities, they are just folds due to specimen preparation. The authors need to be more cautious in writing that gibbositities are present. They should write that gibbositities may be present (given the data they present), but that the highlighted gibbosity could be the result of a cuticle fold due to fossilization and/or cuticle breakage at the most anterior part of the dorsal fold (which they interpret as gibbosity). Therefore, in the definition of the taxon, they have to write that the gibbositities could be present, and not that they are certainly present.

Referee: 1

Comments to the Author(s).

Thank you for making the changes requested in the previous review. This is an intriguing paper that describes a new genus and new species of a tardigrade in Dominican amber, adding to our scant knowledge of fossil tardigrades. I found a few typographical/spelling errors that can be easily corrected. These are listed below:

1. Page 8, Line 25: correct spelling of Hypsibioidea: Hypsibiidae
2. Page 13, Line 17: I think you meant to have a period [.] to end the sentence at "later." Then start a new sentence with "Caribbean...."
3. Page 16, Reference #32: italicize the genus & species of the termite; capitalize Miocene.
4. Page 16, Reference #33: capitalize Paleozoic
5. Page 17, Reference #43: Cambrian Spence Shale Lagerstätte should be capitalized.
6. Page 17, reference #44: There should be a dash rather than a hyphen between tardigrades--adding genes
7. Page 17, Reference: italicize Leptomyrmex
8. Figs 1-4, and Supplementary figures: add comma between "planes" and "respectively"
9. Fig 4 and Suppl Fig 1 and Suppl Fig 2: use a dash instead of a hyphen between darker - more intense and between lighter - less intense
10. Suppl Figs. 3, 4, and 5: correct spelling of Eohypsibioidea

Author's Response to Decision Letter for (RSPB-2021-1760.R0)

See Appendix B.

Decision letter (RSPB-2021-1760.R1)

15-Sep-2021

Dear Mr Mapalo

I am pleased to inform you that your manuscript entitled "A tardigrade in Dominican amber" has been accepted for publication in Proceedings B. Congratulations!!

Data Accessibility section

Open Access

Your article has been estimated as being 9 pages long. Our Production Office will be able to confirm the exact length at proof stage.

Paper charges

Sincerely,

Dr John Hutchinson

Associate Editor:

Board Member

Comments to Author:

The authors have responded to the comments and decided to take the advise to err on caution related to the gibbbsites, which was the main concern from two referees. As such I am happy to let the paper be accepted for publication without any further review.

Appendix A

We are grateful for the valuable comments and suggestions of the different referees. They have significantly helped in the revision of our manuscript. After obtaining new high-quality images, we assigned the fossil into a new genus and species, *Paradoryphoribius chronocaribbeus* gen. et sp. nov. Because of this, we made major revisions by modifying the genus and species description, removing the differential diagnosis between extant *Doryphoribius* species, the morphospace analysis, and community similarity analysis. Additionally, we added phylogenetic analyses to support the taxonomic position of *Pdo. chronocaribbeus* gen. et sp. nov. in the superfamily Isohypsibioidea. All the major changes in the revised manuscript are highlighted in blue in the “tracked version”.

Additionally, the following are our responses to the specific comments and questions that they gave in the original manuscript

REFEREE 1

Page 3

- I am doubtful that this is the correct genus.

→ We have edited this statement in line with our new conclusion of erecting the genus *Paradoryphoribius*.

Page 5

- Insufficient evidence to say equivalently that this is the genus *Doryphoribius*.

→ We have edited this statement in line with our new conclusion of erecting the genus *Paradoryphoribius*

- Isohypsibioidea

→ Edited in the revised manuscript

- In Poiner & Nelson 2019, we described a microinvertebrate with some tardigrade and some mite characters from Dominican amber. We had to trim the amber to obtain a thinner specimen for finer resolution. Could you do that?

→ The images in the original manuscript were already from a trimmed specimen. Due to the position of the inclusion, the amber can no longer be trimmed. Instead, we used a different microscope to obtain higher resolution images of the sample.

Page 6

- Is this the correct format for this reference?

→ Edited in the revised manuscript

- These measurements are only valid if the claws are in the perfect orientation. These claws are not.
- Should also look at other genera, since the genus *Doryphoribius* can not be confirmed.
- add to References: Actual checklist of Tardigrada species (2009-2020, 39th Edition: 17-12-2020) PETER DEGMA 1 ROBERTO BERTOLANI 2 ROBERTO GUIDETTI
- Maximum body size is highly variable. Not a reliable character.
- 6 of these measurements deal with claws and are not independent characters. For any accuracy, you would also need measurements of the buccal-pharyngeal apparatus, which is impossible with your fossil specimen.

→ The morphospace analysis has been removed in line with our new conclusion of erecting the genus *Paradoryphoribius*

Page 7

- Many of these species have been incorrectly reported in the literature, so presence/absence data is inaccurate.

→ Although the species are potentially incorrectly reported, there are currently no literature updating this data that can support the corrections.

- Change to: Richters, 1926; Guil et al. 2019 proposed new class APOTARDIGRADA Guil, Jørgensen & Kristensen, 2019 for the order APOCHELA, suppressed order PARACHELA and replaced it with class EUTARDIGRADA and elevated superfamilies Eohypsibioidea, Hypsibioidea, Isohypsibioidea and Macrobiotidea to orders. Morek et al. 2020b suppressed all these taxonomic acts.}

→ This was an honest mistake by the authors. We already used the correct attribution in the revised manuscript

- Confirmation of the genus is not possible without information on the buccal-pharyngeal apparatus, which is not possible with your specimen.
- *Doryphoribius* species can be divided into four distinct groups using two independent and evident traits: the number of macroplacoids in the pharynx and the presence or absence of cuticular gibbosities (both dorso-lateral and on legs). Michalczyk & Kaczmarek 2010. You do not have these characters identifiable in your fossil specimen.
- and rigid buccal tube with a ...
- Two claw types: the dominant type, with secondary branches being similar in height to the primary branches (all genera with the exception of some *Doryphoribius* spp.); and the second, with secondary branches being clearly shorter than the primary branches (only in some *Doryphoribius* spp.).

→ We have edited this section in line with our new conclusion of erecting the genus *Paradoryphoribius*

- change "swellings" to "protuberances"

→ Edited in the revised manuscript

Page 8

- This is correct. It is in the superfamily Isohypsibioidea, which includes the families Isohypsibiidae and Doryphoribidae.
- probably
- Since you can't see the buccal tube or determine if the ventral lamina is present or absent, you cannot determine the genus!

→ We have edited this section in line with our new conclusion of erecting the genus *Paradoryphoribius*

- In your claw figures, it appears to me that some of the claws have structures that can be interpreted as pseudolunulae.
- not necessarily. Sometimes lunules and pseudolunules can appear to be present on some claws but absent on others.

→ New high-resolution images of the claw do not show any pseudolunules in all claws.

- I disagree. Inadequate sampling may be the reason the other 2 genera have not been reported from the Caribbean.
- No autapomorphies that are visible in the specimen. That doesn't mean they don't have any.
- The genus *Doryphoribius* is polyphyletic and this is clearly visible both from earlier studies (Bertolani et al., 2014a) and the current molecular phylogeny
- I have no problem with the name of the fossil specimen, but insufficient evidence to confirm the name of the genus.

→ We have edited this section in line with our new conclusion of erecting the genus *Paradoryphoribius*

Page 9

- the presence/absence of pseudolunulae is questionable.

→ New high-resolution images of the claw do not show any pseudolunules

Page 10

- Measurements should not be reported if the claws are not in the proper orientation. The numbers are otherwise invalid.

→ Table 1 has been edited in the revised manuscript to only include measurements of claws in the proper orientation

- I do not see this as a cuticular bar. It could be a fold or an artifact.

→ New high-resolution images of the claw show the cuticular bar in between the claws of the right leg IV

- Cuticular bars in Isohypsibids do not connect the bases of the claws.

→ This statement has been edited in the revised manuscript in line with the new high-resolution images obtained

- definitively but appear to be present at the base of the claws in Fig. 3 B and D.

→ New high-resolution images of the claw do not show any pseudolunules

- There are problems with using body length as a distinguishing character. Claw structure is not well-defined in the confocal images
- The gibbosities are not sufficiently well-defined in the fossil specimen to determine their pattern, which is essential in determining the species.

→ We have removed this section in line with our new conclusion of erecting the genus *Paradoryphoribius*

Page 11

- Unless you know the genus is *Doryphoribius*, this section is not valid.

→ We have removed this section in line with our new conclusion of erecting the genus *Paradoryphoribius*

Page 14

- Highly speculative since you can not definitively identify the genus as *Doryphoribius*.

→ We have edited this section in line with our new conclusion of erecting the genus *Paradoryphoribius*

Page 15

- The structure of the claws is not clearly defined with confocal microscopy, as least in this species.

→ New high-resolution images of the claw are presented in the revised manuscript

Page 16

- In some references, the genus is not italicized. Please correct.
- Format for titles of journal references is not consistent. In some cases, the first word and proper nouns are capitalized, but in others, all nouns are capitalized. Please check the format of all references.

→ Edited in the revised manuscript

Page 20

- I can't tell which lumps are gibbosities.

→ New high-resolution images of the dorsal gibbosity are presented in the revised manuscript

- Figures labeled with caps but not in fig legends.

→ Edited in the revised manuscript

Page 21

- Labels are in caps in the figures, but lower case in the legend.

→ Edited in the revised manuscript

- Would X-ray tomography give you better resolution of the claws and cuticle?

→ We are not able to visualize the sampling using X-ray tomography but we were able to obtain high-resolution images using a different confocal microscope for the revised manuscript

Page 22

- pseudolunule?
- pseudolunule?
- not a cuticular bar, in m opinion

→ New high-resolution images of the claw show that pseudolunules are absent while the cuticular bar in between the claws of the right leg IV is present

- A-H Since the figs are labeled with caps, the figure legend should have the same format. Or the figures should have lower case letters to correspond with the lower case in the figure legend. Should be consistent with all the figures. All caps or all lower case--depending on the required format of the journal.

→ Edited in the revised manuscript

- This is not a typical cuticular bar. The bars show up better with phase contrast but may be visible on one side and not the other. Often they don't show up very well. They are also hard to discern with SEM. In my opinion, this is not a cuticular bar.

→ New high-resolution images of the claw show the cuticular bar in between the claws of the right leg IV

Page 23

- This figure is appropriate only if the genus is *Doryphoribius*, which is not definitive in this paper in my opinion.

→ This figure has been edited in line with our new conclusion of erecting the genus *Paradoryphoribius*.

Page 25

- Measurements are invalid unless the claws are in the appropriate orientation

→ Table 1 has been edited in the revised manuscript to only include measurements of claws in the proper orientation

Page 26

- You also have black dots, but no explanation.
- Body length is a highly variable character.

→ This figure has been removed in line with our new conclusion of erecting the genus *Paradoryphoribius*.

Page 27

- D & F look like they may have pseudolunules.

→ New high-resolution images of the claw do not show any pseudolunules

- Claw structure is still not well-defined. Is there another technique that you could use? X-ray tomography? Could you file down the amber specimen even more to get better view? resolution

→ We are not able to visualize the sampling using X-ray tomography but we were able to obtain high-resolution images using a different confocal microscope. Due to the position of the inclusion, the amber can no longer be trimmed.

- Be consistent. Figures labeled with caps but lowercase in fig legend.

→ Edited in the revised manuscript

REFEREE 2

- On p. 7, line 18, indicate the geographic distributions of *Dianea* and *Ursulinius* to allow the reader to better evaluate the likelihood of having been present in the Caribbean in the Miocene but having gone locally extinct.

→ We have deleted this statement in line with our new conclusion of erecting the genus *Paradoryphoribius*.

- The generic assignment draws partly on the presence of dorsal gibbosities. I have to say that these are not tremendously obvious, barely being distinguishable from other areas of the cuticle. Compared to SEM images of gibbosities in extant *Doryphoribius*, I can almost be convinced by the dorsal one on segment 3 in Fig. 1B but I can't really see anything in Figs. 1A and B that has an outline corresponding to the larger, more lateral "gibbosity" drawn in Fig. 1C.

→ We have obtained higher quality images of the dorsal gibbosity which confirmed its presence albeit with a different shape compared to the description in the original manuscript.

Trivial edits:

- P. 3, line 13: indicates that Cedar Lake is in Manitoba.

→ This information is added in the revised manuscript

- P. 3, line 16: "Its precise affinities remain..."

→ Edited in the revised manuscript

- P. 7, line 28: "...dates to the Miocene..."

→ Edited in the revised manuscript

- References 30 and 33: check that "Miocene" and "Carboniferous" are really not capitalised in the titles.

→ Edited in the revised manuscript

REFERREE 3

- Page 4, lines 5-8. The introduction would benefit from being broadened a bit in order to better attract general interest. For the broader audience I would expect at least 4-5 references to recent reviews or overviews on tardigrade morphology, phylogeny and taxonomy, extreme stress tolerance and survival strategies.

→ We acknowledge the benefits of adding more information about tardigrades. However, due to limited word counts, we find it more appropriate to focus more on the rarity of the fossil record of this phylum

- Page 5, Lines 11-13. I do not agree that the limited morphological data presented in this manuscript, i.e. selected claw branch length ratios and a possible presence of dorsal gibbositities (these are not clearly visible from the photos presented) can support the suggested assignment of the fossil to *Doryphoribius*. You simply lack the necessary data on essential buccal-pharyngeal structures in order to do this. The claim that the preserved morphology allows you to erect a new *Doryphoribius* species and assign the fossil to Doryphoribiidae represents an over-interpretation of the data at hand.

→ We have edited this statement in line with our new conclusion of erecting the genus *Paradoryphoribius*.

- Page 5, Line 24-25. In what sense is Tardigrada a "cryptic metazoan phylum"? Please explain or reformulate.

→ Edited in the revised manuscript

- Page 6, Line 20-21: I see 13 (not 10) species listed in your "Supplementary Sheet 2".

→ In line with our new conclusion of erecting the genus *Paradoryphoribius*, we determined that the morphospace analysis is no longer appropriate. Thus, the Supplementary Sheet 2 is no longer used in the revised manuscript

- Page 6, Line 21-23: I suggest that you avoid using statistical terms, such as "mean", when you refer to data from your fossil, where you only have a single data point (n=1).

→ The sentence is already deleted in the revised manuscript

- Page 7, Line 17: Eutardigrada was erected by Richters in 1926. Please provide the correct authority! The text should read: Class Eutardigrada Richters, 1926

→ This was an honest mistake by the authors. We already used the correct attribution in the revised manuscript

- Page 7, Line 26: Yes, the ventral lamina is a defining character of genus *Doryphoribius*, but you don't know whether your specimen has a ventral lamina or not. Accordingly, your data do not provide any support that your specimen actually belongs to this genus.

→ The sentence is already deleted in the revised manuscript in line with our new conclusion of erecting the genus *Paradoryphoribius*.

- Page 8, Lines 1-21: The "Remarks" section should be re-written, clearly acknowledging the fact that you lack defining data on buccal-pharyngeal structures, including (but not limited to) evidence of a ventral lamina on the buccal tube. My advice would be to avoid any over-interpretation of the limited morphological data that you were able to extract from the specimen. I suggest that you put focus on the data at hand, i.e. claw morphology. Along this line, I think that it would be very informative for the reader if you provide a more comprehensive account on isohypsibiid claw structure, in the form of schematic drawings and at the same time provided drawings of the claws on each leg of the fossil specimen.

→ The "Remarks" section has been edited in the revised manuscript in line with our new conclusion of erecting the genus *Paradoryphoribius*. Additionally, schematic drawings of the claws are added.

- Page 9, Lines 1-5: In case data is published on any of these synclusions, please provide references.

→ The data regarding the synclusions has not been published.

- Page 9, Line 8: move abbreviation (AMNH): American Museum of Natural History (AMNH), Division of Invertebrate of Zoology.

→ Edited in the revised manuscript

- Page 9, Line 10: Regarding the "weakly outlined dorsal gibbosities". I am not able to recognize these in Figure 1B. If possible, please provide a magnification of this area of the fossil.

→ New images of the magnified gibbosities are added in the revised manuscript

- Page 9, Line 11: "differing" instead of "differ".

→ Edited in the revised manuscript

- Page 9 & 10, Lines 27-31 & 1-16. At present your description of the claws is rather hard to follow, and I would guess almost impossible to understand for a person that is not familiar with tardigrade claw morphology. Please help the reader and make sure that all details, such as "wide common tract", "triangular claw base", "external",

"internal", "posterior" and "anterior" claws, "primary" and "secondary" branches etc. are clearly indicated on Figures 3A-H (e.g. with arrows and/or abbreviations). Please also see my suggestion above regarding schematic drawings of the claws on each leg – I think this would really improve the readability of your manuscript.

→ Schematic drawings of the claws and labels of the claw parts are added in the revised manuscript

- Page 10, Line 20: Regarding "claw structure (claw pairs slightly different from each other and each claw has br <70%)". According to Table 1 the claim that " br <70%" does not apply to all claw pairs. Please explain.

→ Table 1 has been significantly updated in the revised manuscript and this statement has been deleted

- Page 10, Lines 21-26: this represents an over-interpretation of your data.

→ The sentences are already deleted in the revised manuscript in line with our new conclusion of erecting the genus *Paradoryphoribius*.

- Pages 10, 11 & 12, Lines 28-30, 1-31 & 1-11. I do not agree that the limited morphological data available from this fossil can support the suggested assignment to *Doryphoribius* (see my comments above). Accordingly, I suggest that you leave out the comparison to extant *Doryphoribius* species.

→ The sentences are already deleted in the revised manuscript in line with our new conclusion of erecting the genus *Paradoryphoribius*.

- Page 14, Lines 1-5: As I do not agree with the suggested assignment to *Doryphoribius*, I would also suggest that you delete or reformulate these two sentences.

→ The sentences have been edited in the revised manuscript in line with our new conclusion of erecting the genus *Paradoryphoribius*

- Figure 1: Please provide a magnification of the area with gibbosities.

→ New images of the magnified gibbosities are added in the revised manuscript

- Figure 3: Insert labels to mark details, such as "common tract", "triangular base", "external" and "internal" claws, "primary" and "secondary" branches etc. Also please provide drawings of the claws on each leg.

→ Schematic drawings of the claws and labels of the claw parts are added in the revised manuscript

- Figure 5: Consider also inserting the heterotardigrade fossil that was found together with *Beorn leggi* on this figure.

→ The heterotardigrade affinity of this fossil has not been substantially proven and thus, we opt not to include it in the figure

- Table 1. Given the difficulties in obtaining accurate measurement, I would leave out decimals in the calculated br values. To ease readability please explain "br" in table text.

→ Edited in the revised manuscript

- Supplementary Figure 1: suggest you avoid using the term "mean", when you refer to the length of your fossil.

→ The figure has been removed in the revised manuscript

- Supplementary Figure 2: delete the repetition "under confocal microscope with autofluorescence at 488 nm" (Lines 3-4).

→ Edited in the revised manuscript

Appendix B

We are grateful for the valuable comments and suggestions of the different referees. All the major changes in the revised manuscript are highlighted in blue in the “tracked version”.

Additionally, the following are our responses to the specific comments and questions that they gave in the manuscript

REFEREE: 2

- *Just one suggestion. I don't see the point of Supplementary Figure 3. Figure 5 shows the G/C values for the reasonably supported nodes in this implied weighted tree. All Supplementary Figure 5 adds is a few nodes that have G/C values of 3, 4, 6 and 47. These poorly supported nodes really don't warrant reporting.*

→ We appreciate the reviewer's comments and agree that the information content in Supplementary Figure 3 is limited to additional support values. These nodes are ultimately not the focus of the paper as they are outside of the monophyletic isohypsiboid clade we recover with relatively high support. However, in the interest of transparency we believe it is best to include these support values, even if they are low, to allow readers to assess relationships we recovered with this dataset in the future.

REFEREE: 4

- The authors need to be more cautious in writing that gibbosities are present. They should write that gibbosities may be present (given the data they present), but that the highlighted gibbosity could be the result of a cuticle fold due to fossilization and/or cuticle breakage at the most anterior part of the dorsal fold (which they interpret as gibbosity). Therefore, in the definition of the taxon, they have to write that the gibbosities could be present, and not that they are certainly present.

→ Cuticular gibbosities are weakly outlined or poorly visible in some tardigrade species (i.e. *Doryphoribius dawkinsi*, *Doryphoribius mariae*; see: DOI:10.5281/zenodo.193905). This fossil sample also appears to have this type of gibbosity which could explain why this structure is not very visible. Furthermore, given the height of this cuticular structure and its contour (based on the CLSM reconstruction in Figure 4A), we believe that it could be interpreted as a gibbosity. However, we concede to the referee that this structure can also be a cuticular fold resulting from the fossilization process. Since no other gibbosity-like structures can be found in the fossil, it is currently impossible to distinguish whether the structure is a gibbosity or a cuticular fold. Additionally, there are no taphonomic studies on how tardigrade gibbosities are affected during decay or how cuticular folds are made during fossilization that can help in deciphering the identity of this cuticular structure. Given all these, we decided to change our interpretation in the “Genus Diagnosis” of the gibbosities from “**present**” to “**may be present**” to reflect this uncertainty. We also added the following line: “**This structure could also be a cuticular fold caused by the fossilization**”

process” in the “Species Description” to give other possible interpretations of this structure.

Because of this uncertainty in the character, we had to change the coding of character 2 (Dorsal cuticle protuberances) of *Paradoryphoribius chronocaribbeus* from **“1”** to **“?”** and to redo all the phylogenetic analyses. This coding change, however, did not produce any significant changes with the results (i.e., same tree topologies, no significant difference in the branch support, *Pdo. chronocaribbeus* still cluster with other isohypsibioids). Figures 5 and Supplementary Figures 3-5 were changed to represent the results obtained from these new analyses. In the schematic drawing in Figure 1C, the label was also changed from **“gibbosity”** to **“gibbosity?”** to reflect the uncertainty of the nature of this cuticular structure. The description in Figure 4 was also changed from **“dorsal gibbosity”** to **“dorsal gibbosity-like structure”**

Ultimately, the presence or absence of gibbosities does not alter the major conclusions of our paper (i.e., erection of a new genus and species) since the fossil show a significantly different morphology of the buccal apparatus from other tardigrades.

REFEREE: 1

- I found a few typographical/spelling errors that can be easily corrected. These are listed below:
 1. Page 8, Line 25: correct spelling of Hypsibioidea: Hypsibiidae
 2. Page 13, Line 17: I think you meant to have a period [.] to end the sentence at "later." Then start a new sentence with "Caribbean...."
 3. Page 16, Reference #32: italicize the genus & species of the termite; capitalize Miocene.
 4. Page 16, Reference #33: capitalize Paleozoic
 5. Page 17, Reference #43: Cambrian Spence Shale Lagerstätte should be capitalized.
 6. Page 17, reference #44: There should be a dash rather than a hyphen between tardigrades--adding genes
 7. Page 17, Reference: italicize *Leptomymex*
 8. Figs 1-4, and Supplementary figures: add comma between "planes" and "respectively"
 9. Fig 4 and Suppl Fig 1 and Suppl Fig 2: use a dash instead of a hyphen between darker - more intense and between lighter - less intense
 10. Suppl Figs. 3, 4, and 5: correct spelling of Eohypsibioidea

→ All the typographical errors have been corrected in the revised manuscript.